# Development of an Automated Minimum Foot Clearance Measurement System: Proof of Principle

**DOI:** 10.3390/s21030976

**Published:** 2021-02-02

**Authors:** Ghazaleh Delfi, Megan Kamachi, Tilak Dutta

**Affiliations:** 1KITE—Toronto Rehabilitation Institute, University Health Network, Toronto, ON M5G 2A2, Canada; megan.kamachi@mail.utoronto.ca (M.K.); tilak.dutta@utoronto.ca (T.D.); 2Institute of Biomedical Engineering, University of Toronto, Toronto, ON M5S 1A1, Canada

**Keywords:** computer vision, falls, gait, marker-less gait analysis, minimum foot clearance, motion capture, trips

## Abstract

Over half of older adult falls are caused by tripping. Many of these trips are likely due to obstacles present on walkways that put older adults or other individuals with low foot clearance at risk. Yet, Minimum Foot Clearance (MFC) values have not been measured in real-world settings and existing methods make it difficult to do so. In this paper, we present the Minimum Foot Clearance Estimation (MFCE) system that includes a device for collecting calibrated video data from pedestrians on outdoor walkways and a computer vision algorithm for estimating MFC values for these individuals. This system is designed to be positioned at ground level next to a walkway to efficiently collect sagittal plane videos of many pedestrians’ feet, which is then processed offline to obtain MFC estimates. Five-hundred frames of video data collected from 50 different pedestrians was used to train (370 frames) and test (130 frames) a convolutional neural network. Finally, data from 10 pedestrians was analyzed manually by three raters and compared to the results of the network. The footwear detection network had an Intersection over Union of 85% and was able to find the bottom of a segmented shoe with a 3-pixel average error. Root Mean Squared (RMS) errors for the manual and automated methods for estimating MFC values were 2.32 mm, and 3.70 mm, respectively. Future work will compare the accuracy of the MFCE system to a gold standard motion capture system and the system will be used to estimate the distribution of MFC values for the population.

## 1. Introduction

Falls are a major health issue, especially among those who are 65 or older, for whom falling is the most common reason for non-fatal injuries [1]. Each year, 3 million people in the United States suffer injuries caused by falls [2]. Twenty percent of these falls lead to head injuries or fractures that can trigger a sudden downward spiral in health [3,4]. It is estimated that just over half of falls in older adults are caused by tripping [5] that occurs when an individual fails to adjust their gait when negotiating obstacles and raised surfaces.

A trip typically occurs during swing phase at or near the Minimum Foot Clearance (MFC) point if the trajectory of the foot is suddenly interrupted by an obstacle [6]. The MFC occurs at the point in time when the distance between the lowest point on the shoe and the ground reaches a local minimum. A trip occurring at this point is likely to lead to a loss of balance and therefore a subsequent fall [7,8] because the feet are close together and the base of support is small. Combined with the fact that the foot has a relatively high velocity, the potential for a trip-related fall is considered to be the highest at MFC [7]. Therefore, individuals with lower or more variable MFC values are at higher risk of trip-related falls [9].

Therefore, a better understanding of the distribution of MFC values for the overall population would allow for safer design of the built environment. Current guidelines vary considerably from one jurisdiction to the next. For instance, the Americans with Disabilities Act (ADA) [10], vertical changes in level on walkways are allowed to be a maximum of 6.4 mm while the city of Toronto [11] allows level changes up to 13 mm. Existing research demonstrates that 13% of the older adults have MFC values below 6mm and that even healthy young adults would trip on an unseen 5 mm obstacle 1 in every 95 strides [12].

Conditions such as advanced age, Parkinson’s disease, performing dual tasks, or simply being fatigued, can cause individuals to have reduced mean MFC mean or increased MFC variance compared to young healthy individuals.

### 1.1. MFC Measurement

There is a large body of literature when it comes to the measurement of MFC values but nearly all perform the measurement in the laboratory and most use optical motion capture [6]. Optical motion capture systems include a network of infrared video cameras and reflective markers that are placed on bony landmarks on participants’ feet to be able to track movement. The processes for both data collection with a motion capture system (camera calibration, marker placement) and processing (labeling and filling in gaps in marker trajectories) can be time consuming and the amount of equipment involved makes it difficult to use these systems outside of the laboratory environment. These challenges also tend to limit the number of individuals that are able to participate in studies performed in controlled laboratory environments and make it difficult to establish a population-level MFC distribution. It is also important to note that MFC values can be overestimated by 6.5–16.2% when measured in the lab compared to real-world settings [13].

Therefore, new methods for efficiently measuring MFC in the real-world from large numbers of individuals are needed to develop a better understanding of MFC values for the population.

### 1.2. Objective

The objective of this work is to present the Proof of Principle for an automated Minimum Foot Clearance Estimation (MFCE) system for estimating MFC values to address the limitations of existing MFC estimation methods. The MFCE system was designed to quickly gather and process MFC estimates for large numbers (thousands) of pedestrians on outdoor walkways that include level ground as well as obstacles of different sizes. This data will then be used to create population level distributions of MFC values that we think will be beneficial for revising standards for the design and maintenance of outdoor walkways. We describe our data collection and analysis methods (both manual and automated) and describe the development and evaluation of the automated footwear detection, trajectory estimation and laser detection subsystems. Finally, we compare the manual and automated analysis methods and comment on the potential for the system to be used for estimating the distribution of MFC values for the population.

## 2. Methods

The MFCE system includes two modules: A data collection module and an automated data analysis module that processes video data offline. The benefits of this novel system are that:It can easily collect large amounts of data quickly from many pedestriansThe measured MFC values will represent the lowest point of the entire footThe system is marker-less, there is no need for attaching markers to the participants or control the environment in any wayIt utilizes a simple and relatively low-cost equipment (consumer grade video camera and two lasers)The pedestrians are not aware of the device recording them therefore this measurement will not affect their walking performance

The overarching objective of this work is to develop a system to define the population-level MFC distribution for pedestrians walking over level-ground walkways as well as over walkways that include a range of imperfections including surface discontinuities, obstacles or bumps. The resulting foot clearance distribution data will contribute to developing evidence-based guidelines for outdoor walkway maintenance and inform other strategies for minimizing the risk of trip-related falls on outdoor walkways.

### 2.1. Data Collection Module

The data collection module (Figure 1) consists of a consumer-grade video camera (Sony FDR AX33) and two calibrated parallel laser beams (Galileo pro, 5 mW, Laserglow Technologies) positioned within the field of view of the video camera such that they are projected onto the pedestrian’s lower leg. The videos recorded by the camera are used to track the foot trajectory of the pedestrian and extract the MFC value. The known distance between the two laser beams is used to define a scale on the video image so that foot clearance distances measured in pixels on the video image can be converted to distances in mm.

The device was designed to be positioned at ground level next to a public walkway to collect sagittal plane video images of pedestrians’ feet and lower legs as they walk by (Figure 2). Note that the pedestrians were not aware of the fact that they were being recorded. Therefore, we were unable to present demographic/anthropometric information or spatiotemporal gait parameters or allow for repeated measures with any given pedestrian. To setup the system, the centre of the camera lens is aligned with the ground plane such that its view is parallel to the ground plane. Furthermore, the lasers are precisely aligned to be parallel to each other, so that the distance between the two laser-beams remains constant regardless of distance to the system. Figure 3 shows the distance each pixel represents for subjects located in different distances from the camera. These calculations assume that the video is recorded in HD resolution (1080 by 1920 pixels). It is apparent that the further away the subject is from the camera, the less accurate measurements derived from the image will be. Based on the average width of sidewalks in Toronto, we assumed that the pedestrians will be maximum two meters away from the camera. Figure 3 shows that the accuracy of our system at this distance is 1.59 mm.

Video was collected using the following camera settings: resolution of 1080 by 1920 pixels, frame rate of 60 p, shutter speed set to 1/10,000 s, and bit rate settings 130 Mb/s. This study was approved by the University Health Network Research Ethics Board and the University of Toronto Research Ethics Board.

### 2.2. Data Analysis

In this section we present the manual method for estimating MFC values from the video captured using the Data Collection Module and describe the development of the automated Data Analysis Module along with our method for comparing the manual and automated analyses.

#### 2.2.1. Manual Analysis of MFCE Video Data

To manually measure MFC values from MFCE video data, separate video clips were cropped for each pedestrian (50 frames) and identified the frame(s) where the laser beams land on their lower leg/foot (Figure 4. These frames were then viewed using MB Ruler, which allowed extracting precise measurements of the distance between the two lasers in pixels by zooming in to see individual pixels making up the video images. The pixel-to-millimeter conversion factor (*k*) was calculated using Equation (Equation 1) where the known distance between the two parallel laser pointers (164 mm) was divided by the distance in pixels (*x*).
(1)k=164/x.

Next, we measured the distance (in pixels) between the lowest point of the pedestrian’s footwear and the ground plane for each frame in where the foot was in swing phase. These distances were plotted for all frames from each pedestrian and the local minima was extracted. We measured this distance (*d*) and calculated MFC value as shown in Equation (Equation 2).
(2)MFC=d×k.

#### 2.2.2. Automated Analysis of MFCE Video Data

Automated analysis of MFCE video data requires five steps: selecting the video frames where each pedestrian is visible, detecting the ground plane, detecting the footwear, finding the MFC point in the swing foot trajectory, and locating the laser dots in the video image. In this paper, we describe how we used computer vision and machine learning techniques to automate three of these steps: footwear detection, estimation of the MFC point in the swing foot trajectory, and laser dot location detection. Future work will focus on automating the remaining two steps: selecting the video frames for each pedestrian, and ground plane detection.

**Footwear Detection.** Mask Region-based Convolutional Neural Network (Mask R-CNN) [14] was used for instance segmentation to locate the location and boundaries of the shoes. Mask R-CNN expands on the previously known Faster-RCNN [15], which is a method for finding bounding boxes and adds a layer for finding an exact mask (outline) for objects present in the input image. A neural network was trained to search for footwear in MFCE video images and return their locations with pixel-wise accuracy using Matterport’s implementation of Mask R-CNN [16] with a backbone of ResNet101. Table 1 shows the configuration of the trained network.

Our data set consisted of 500 frames taken from 10 video frames from each of 50 pedestrians walking through the camera view from three different locations. All three locations were in the vicinity of Toronto Rehabilitation Institute and were sections of level concrete sidewalk with not observable discontinuities or obstacles. The three sites were similar in appearance except that the lighting conditions varied (locations A and B were in shade while location C was in bright sun). Eighteen pedestrians were captured location A, 19 in location B and 13 in location C. We included 370 frames from 37 different pedestrians (locations A and B) to train the network and the remaining 130 frames from 13 pedestrians (location C) to test the performance of the network. The footwear visible in each frame was manually traced for each frame as shown in Figure 5.

Each foot’s distance from the sidewalk was determined for each frame. The foot that was further from the sidewalk was labelled the swing foot and the other was labelled the stance foot. In video frames where the two feet were overlapping, the network initially struggled to find the exact boundaries of the feet. To solve this problem, we used knowledge of the number of footwear masks found in previously analyzed frames to categorize each new frame of video into one of four states by following the state diagram shown in Figure 6. Each frame was categorized as either ‘No one’: no pedestrians in the field of view of the camera, ‘Begin’: A person has just entered the frame and only foot is visible, ‘Fully visible’: both feet are visible and distinguishable by the network, and ‘Overlap’: two feet are overlapping in the sagittal plane and not separable by the network. This categorization allowed the system to easily identify frames where there were two feet overlapping in an image which caused an occlusion flag to be raised. When this flag was raised, swing foot data was retrieved from the previous frame. Then, the current frame was searched to find the most similar patch to the swing shoe. Then the swing shoe was rotated, and the search was performed again. We compared the similarity between the bottom edge of the detected swing shoe with edges present in the overlap frame to find the patch that best aligned with the bottom of the mask in the search image. The shoe was rotated 20 degrees clockwise and 20 degrees counter-clockwise, in steps of 5 degrees. The patch with the highest similarity was chosen to show the location of the swing foot in the current frame.

We evaluated the performance of the Footwear Detection System by calculating the Intersection over Union (IOU) between the detected footwear masks and the ground truth masks. We also calculated the RMS error between the bottom of the detected footwear mask and the ground truth masks.

**Finding the MFC Point in the Swing Foot Trajectory.** To calculate the trajectory, we needed to first separate individual strides since most pedestrians walked an average of 2.5 strides within the MFCE field of view. To separate strides, we identified the direction each person was moving based on which side of the video image their shoe was detected in first. Next, we tracked the position of the center of the swing foot’s mask. The change in direction of the swing foot was used to indicate the transition to stance phase indicating the end of the stride.

Next, the lowest point of the swing foot (foot clearance) was located in each frame and stored in a vector and a 5th degree polynomial was fit to the set of points making up a stride. A 5th degree curve was chosen based on previous work that had identified foot trajectory has three inflection points [17]. Figure 7 shows an example of the trajectory formed by connecting the lowest point found on the swing foot in stride along with the 5th degree polynomial fit to those points.

**Laser Dot Location Detection.** The following assumptions were made about the characteristics of the lasers in the video images to locate them:The laser dots were among the brightest points in the entire imageThe two lasers were aligned nearly vertically within the 2D video imageThe laser points were circular

The lasers were located by thresholding [18] to eliminate the darker pixels in the images and performing blob detection [18] to find two bright, circular groupings of pixels vertically aligned in the images. Part of our automated system included a user interaction to confirm whether the lasers were indeed visible on the pedestrian’s leg in the cases where the algorithm had failed to locate the two laser dots. This functionality was added to address the 10% of cases where the algorithm failed to find the laser points in the video images because the pedestrian was walking fast enough that no video frames included both lasers were projected onto the pedestrian’s leg/foot. If both lasers were found on the pedestrian’s legs/feet, the user was asked to right click on the centers of the lasers.

### 2.3. Comparison of Manual and Automated Methods

To evaluate the performance of the automated system, we compared the RMS errors of automatic and manual MFC estimates from 10 pedestrians from Site A. The RMS errors were calculated by comparing to the mean manual measurement in each case.

Five manual raters (3 female and 2 male, age 28 ± 8 years) were asked to estimate MFC values from 10 pedestrians recorded with the MFCE system. Each manual rater was asked to:Locate the laser points: The raters were asked to record the frame number in each video clip where the laser points were projected onto each pedestrian’s lower leg or foot and click the centre of each to record their 2D coordinates in pixels using MB Ruler.Locate the ground plane: The raters were asked to define the ground plane in each video clip by clicking on two points to record their coordinates.Locate the frame where the MFC occurred and estimate the distance to the ground plane that was defined previously.

The raters were given two additional video frames for each pedestrian to analyze to allow for determining the magnitude of each potential source of error present in the manual analysis. The first was the video frame for each pedestrian was pre-determined to be the one in which the MFC point occurred and the raters were asked to define the ground plane and measure the distance between the lowest point of the shoe and the ground.

The second of these two additional frames included the same frame used in the previous task, but with the ground plane pre-defined as well. This meant raters only needed to select the lowest point on the pedestrian’s shoe to estimate the MFC values for each of the 10 pedestrians again.

The results of the manual methods were compared to the results obtained from using the automated sub-systems to analyze the video data from the same pedestrians. The videos were manually cropped and the ground plane was manually defined as part of the automated process.

## 3. Results and Discussion

### 3.1. Performance of the Three Automated Subsystems

#### 3.1.1. Footwear Detection

The footwear detection network was found to have an Intersection over Union of 85%. The combined approach consisting of the network and the post-processing occlusion method to locate the shoes in a stride was able to identify the bottom of the swing shoe with an average RMS error of 6.8 pixels or 5 mm compared to the manual ground truth masks.

#### 3.1.2. Finding the MFC Point in the Swing Foot Trajectory

The network was able to correctly distinguish between the stance foot and the swing foot in 100% of the cases of which the two feet were not overlapping in the sagittal plane. When they are overlapping, the post-processing algorithm is designed to only look for the swing shoe.

#### 3.1.3. Laser Dot Location Detection

Our algorithm was able to locate the laser pointer dots in each frame in 60% of the cases initially without the need for user interaction. The use of user input was able to locate the laser points in the remaining 40% of frames where the laser dots were visible on the pedestrian’s leg/foot.

### 3.2. Comparison of Manual and Automated Methods

The Bland-Altman plot in Figure 8 shows the inter-rater agreement for the five raters’ manual MFC estimates from the 10 pedestrians. The overall RMS error of these estimates was 2.32 mm.

The automated algorithm was also used with the same 10 pedestrians to see if the automated measurements’ RMS was comparable to that of the manual method.

Figure 9 shows the mean and standard deviation of manual measurements for each pedestrian (blue box), as well as marking the value reported by the automated system for each pedestrian (green cross). The RMS error for the MFC estimates from automated system was 3.70 mm.

The manual evaluation demonstrated that there was a wide variability in MFC estimates for different manual raters. If we eliminate the conversion factor and only compare the measurements done in pixels, the RMS error from the mean of all manual raters was 2.88 pixels.

After controlling for the chosen MFC video frame, the average root mean squared error for different raters dropped to 2.36 pixels (0.5 pixel less than the first step).

After pre-defining the ground plane and pre-determining the MFC frame, the average RMS error for different raters further dropped to 0.97 pixels (1.11 pixels less than the second step).

Table 2 shows the manual measurements mean and standard deviation, as well as RMS errors for the manual and automated measurements for each pedestrian.

The results demonstrate that the performance of the automated elements of the MFCE system are comparable to the manual measurements and are therefore an acceptable substitute for the analysis of data gathered with the MFCE device. However, there is still room for improving the performance of the algorithm and the overall performance of the device.

### 3.3. Limitations

#### 3.3.1. Systematic Errors

The following are the potential sources of random and systematic error in the MFCE system. (1) Video camera image resolution: Higher resolution (4K) consumer-grade cameras are available and would increase accuracy of the system. (2) Camera to sidewalk alignment: If the sidewalk was at an angle relative to the camera, it would become more difficult to define the ground plane accurately and this variability will result in uncertainty in the location of the MFC values. (3) Pedestrian route to camera angle: To correctly measure the MFC using our settings, we assume that the pedestrians are walking in a plane perpendicular to the camera. However, this is not always the case. (4) Lasers not being parallel: If the lasers were not parallel, the distance would not stay the same throughout the depth of the sidewalk location and the measured pixel-to-millimeter scale would be wrong.

#### 3.3.2. Validation

This system is yet to be validated against the gold standard to evaluate the performance. A future study will compare MFC estimates produced by our system and an optical motion capture system.

### 3.4. Future Work

Future work will include the use of higher resolution (4K) video recording, automating video cropping, ground plane detection, and measuring the accuracy of the system by comparing to a gold standard optical motion capture system. The device will be used to collect data to gather MFC data from a wide range of public walkways and pedestrians. Data will be gathered from approximately 1000 members of the public to estimate their MFC values using walkways that are level, sloped, and locations with existing tripping hazards of different sizes to determine if and how people adjust their foot clearances. These findings will be used as the basis for influencing positive changes to minimize outdoor trips, that is, influencing policy changes for outdoor walkway maintenance. Future work will also assess the potential benefits and limitations of using the MFCE system in the clinical environment, in the home, or in other settings.

## Figures and Tables

**Figure 1 sensors-21-00976-f001:**
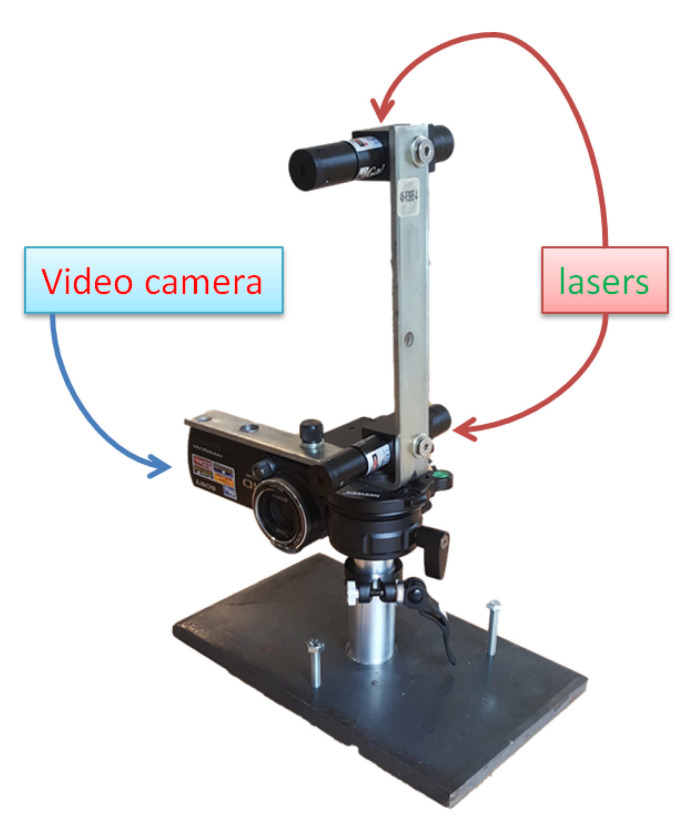
This image shows the Minimum Foot Clearance Estimation (MFCE) device. The device is made of a single video camera and two parallel laser pointers positioned on top of the camera. This device is meant to be placed on outdoor walkways to gather video of the feet and lower legs of pedestrians (general public) walking by. The camera is used to gather videos of passerby’s feet to assess their foot clearance. The parallel lasers have a fixed distance that acts as a reference scale in the recorded footage.

**Figure 2 sensors-21-00976-f002:**
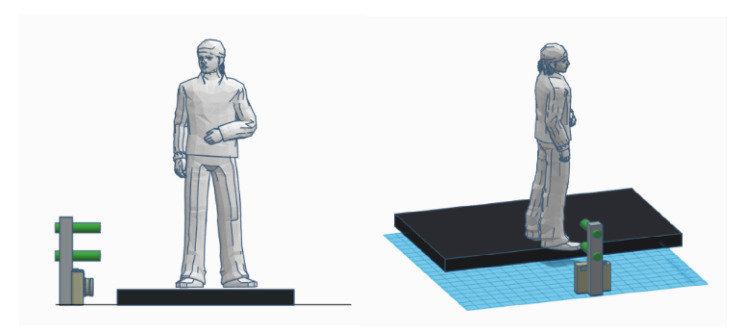
This image shows the MFCE device setup on the sidewalk. The device is placed on the street facing the sidewalk, with the center of the camera aligned to the sidewalk plane. The system will record passerby gait information and analyze the foot clearance automatically. No additional set-up is required.

**Figure 3 sensors-21-00976-f003:**
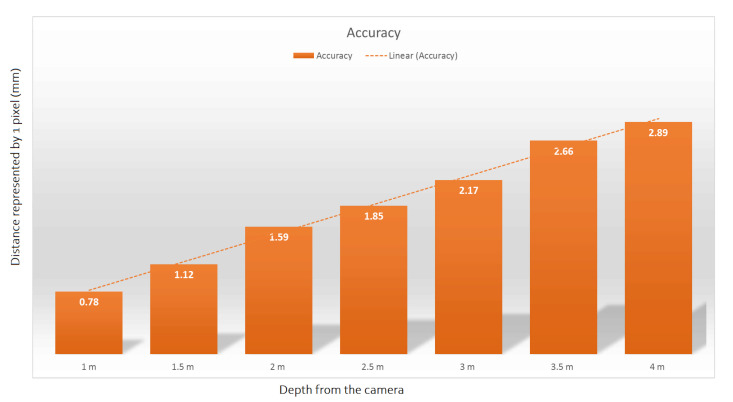
This figure shows how much real-world distance (mm) is represented by each pixel for different camera-to-subject distances. Note that this is assuming that the camera has HD resolution (which was the case in our data collection). The accuracy of the measurements rely on how close the subject is to the camera. The farther away the subject, the more potential error in estimating the distances in the image.

**Figure 4 sensors-21-00976-f004:**
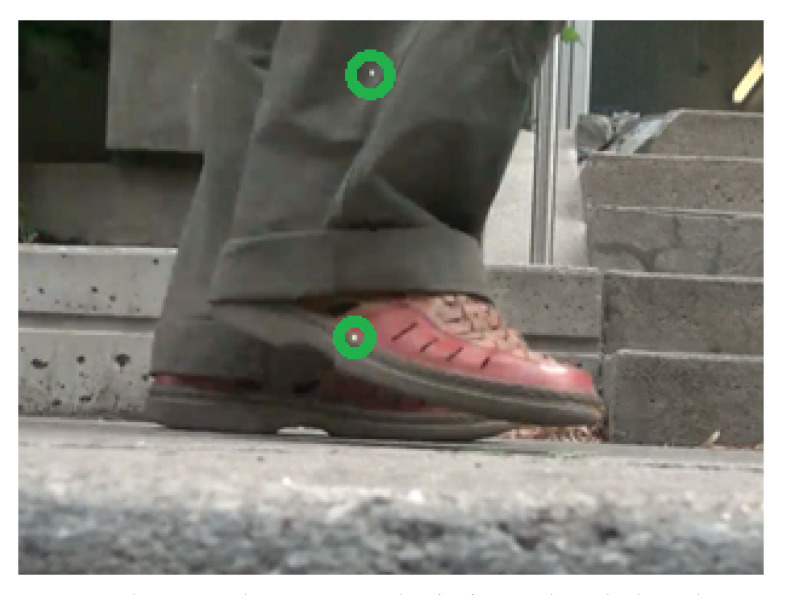
This image shows an example of a frame where the laser dots are projected onto the pedestrian’s foot. The laser dots are marked with a green circle. The distance between the center of these two dots in the picture is equal to the fixed distance between the two laser pointers on the device.

**Figure 5 sensors-21-00976-f005:**
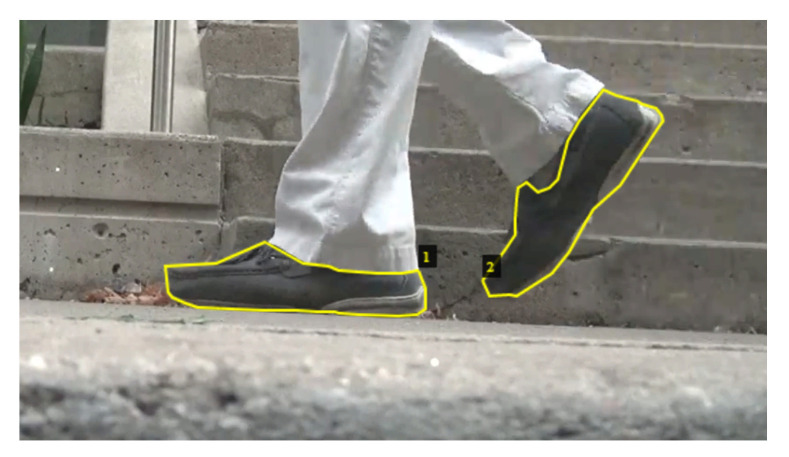
This image provides an example labeled frame. The two instances of footwear are traced with a yellow line. The masks are then given to the network as the ground truth.

**Figure 6 sensors-21-00976-f006:**
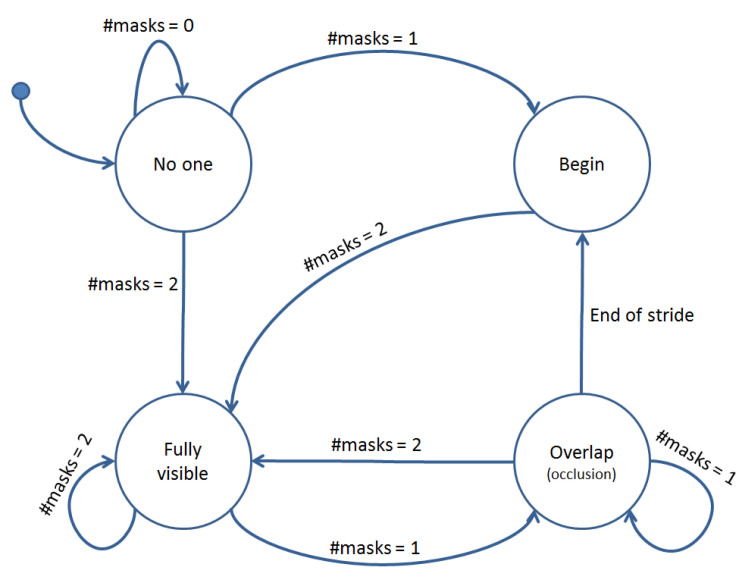
This figure provides the state diagram of the possible states of each frame in the captured pedestrian videos.

**Figure 7 sensors-21-00976-f007:**
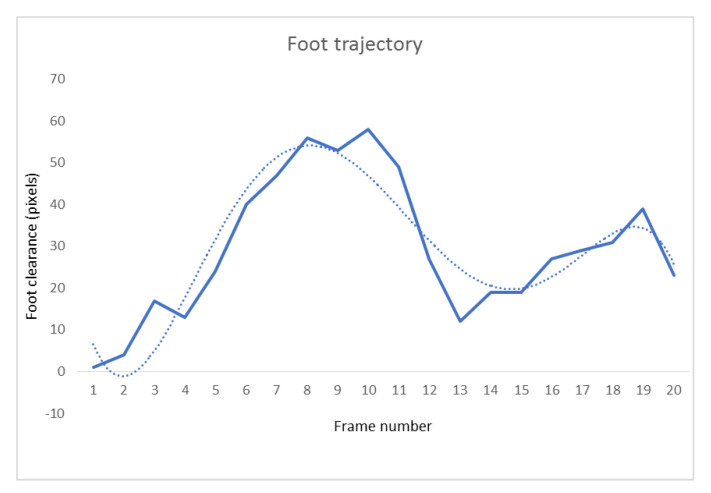
This figure shows the trajectory of the swing foot in one stride; the solid line shows the trajectory based on the lowest point on the detected swing masks and the dotted line shows the fitted 5th degree curve.

**Figure 8 sensors-21-00976-f008:**
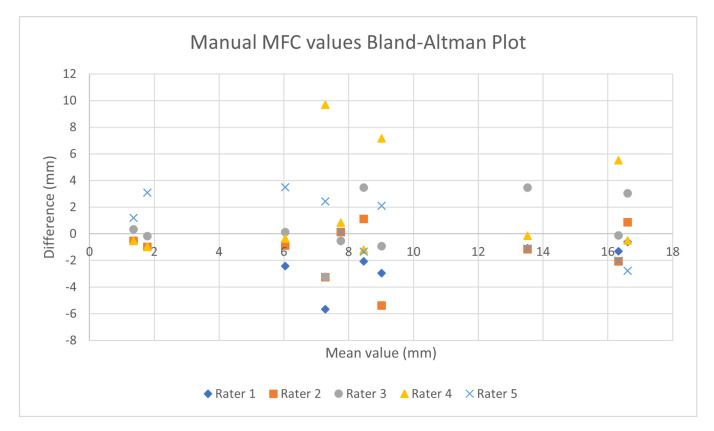
The Bland-Altman plot showing the inter-rater agreement between the five raters in the final evaluation.

**Figure 9 sensors-21-00976-f009:**
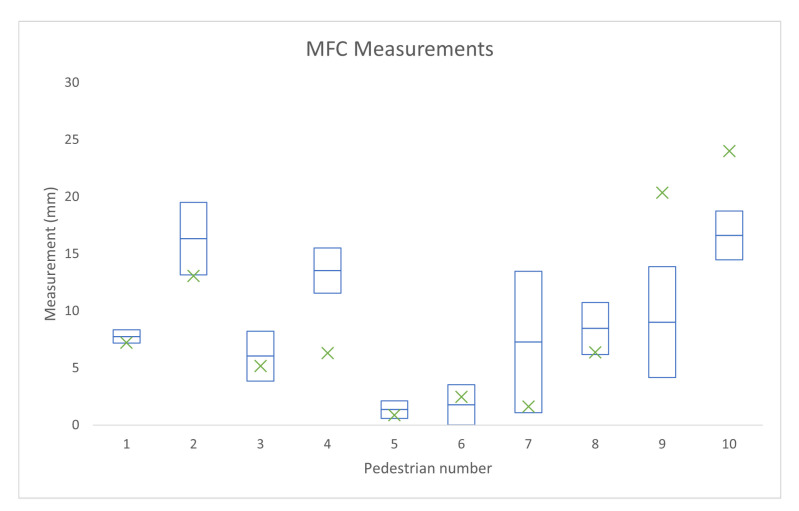
The Manual measurements’ range within one standard deviation of each pedestrian’s respective mean is shown with the blue boxes and the automatic measurement for each case is marked by the green x.

**Table 1 sensors-21-00976-t001:** Mask-RCNN network configuration.

Parameter	Value
GPU count	1
Images per GPU	2
Steps per epoch	1000
Validation step	50
Learning rate	0.001
Weight decay	0.0001
Back bone	ResNet101
FPN layers’ size	1024
RPN NMS threshold	0.7
Number of classes	1
Anchors per image	256
Detection minimum confidence	0.7

**Table 2 sensors-21-00976-t002:** The manual measurements’ mean and standard deviation and the manual and automatic measurements’ root mean squared error for Minimum Foot Clearance (MFC) measurement (mm) for each of the pedestrians.

MFC Values (mm)										
	**1**	**2**	**3**	**4**	**5**	**6**	**7**	**8**	**9**	**10**
Manual mean	7.76	16.33	6.04	13.52	1.36	1.79	7.28	8.46	9.89	16.61
Manual STD	0.57	3.19	2.18	1.98	0.77	1.76	6.19	2.28	5.04	2.14
Manual RMS	0.51	2.85	1.95	1.77	0.69	5.54	1.58	2.04	4.34	1.91
Auto RMS	0.56	3.26	0.86	7.23	0.50	5.66	0.67	2.09	11.34	7.40

## Data Availability

The data used in this study can be found at https://dataverse.scholarsportal.info/dataset.xhtml?persistentId=doi:10.5683/SP2/CJDVLM.

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
