# Peer review of "Development of an Automated Minimum Foot Clearance Measurement System: Proof of Principle"

_sensors, 2021, doi:10.3390/s21030976_

Round 1

Reviewer 1 Report

The paper addresses an interesting topic; foot clearance is a relevant indicator of the gait quality and provides information about the risk of falls in older adults and other related-gait pathologies. The authors propose the measurement of the MFC (Minimum Foot Indicator). Fifty users were recorded during the walking with a video camera and two laser pointers (for calibration purposes). Two offline methods for the estimation are presented and compared to each other. On the one hand, an automatic method for footwear detection and the MFC in the swing face trajectory.

On the other hand, a manual method was performed by five raters. The error between raters and the error with the automatic sample were presented. This last method was tested with recordings from 10 users. There is missing information about formulation and practical information. I also have some concerns about the impact of this work, that I can summarize in the following points:

  1. The introduction presents the relevance of the MFC measurement, but it is not defined how the clinicians measure this indicator. It is required to compare the perspective of the clinicians with the proposed method. Indeed, it is unclear how this proposal will be used in clinical settings. One of the main issues of this work is the lack of clinicians’ measurements to compare the estimated measurements.
  2. The Data Collection module is barely presented. The authors should provide a precise setup of the measurements and the variables and parameters that define the acquisition. It is also unclear how the distance from the user to the module affects the quality of the measures. This factor should be validated in the experimental section.
  3. The user’s data is not defined. Anthropometric information of the pedestrians and spatiotemporal gait parameters must be clearly presented. These factors should be analyzed in the discussion section and explain how those parameters affect the measurements. Another main issue of this work is the lack of testing different speeds with the same users to evaluate the methods' performance.
  4. The figures, in general, present vague information. The authors should improve the quality and information of the figures. For instance, figures 1 to 5 do not represent any relevant information.

I invite the authors to address all points presented to evaluate this work properly. At this point, the lack of evaluation with a motion capture system reduces this work's impact, and I would argue to reject this work.

Reviewer 2 Report

Authors proposed an innovative system for foot clearance based on videocamera and neural network for footwear recognition. Procedure has been tested with 18 pedestrians for a total of 500 video frames. Training and test procedure have been applied for algorithm validation. The paper is generally well written and the topic is interesting.

I would like to suggest some modifications to increase the scientific soundness of the paper.

  1. Authors should report more details, if any, about location A, B, C (different terrains?).
  2. Author should report demographical data of pedestrians involved in the study.
  3. Algorithm performance should be assessed also by other syntetic indices: recall, precision, F1 score etc.
  4. Authors should justify the choice of divided training and testset without performing cross validation, that it has been demostrated as the most robust methodology for the AI algorithm training.
  5. Authors should stress the possible applications of such methodology in real life. Is it possible in domotic applications? It is only for clinical use? I wonder to receive more information on it.

Round 2

Reviewer 1 Report

Dear Editor,

I consider that this work has been improved, and it should be accepted for publication.

Best regards,

Carlos

Reviewer 2 Report

Authors solved issues I have highlighted before.